# COVID-19 vaccination uptake and determinants of booster vaccination among persons who inject drugs in New York City

**Mehrdad Khezri**[1,2]*, **Courtney McKnight**[1,3], **Chenziheng Allen Weng**[1],
**Sarah Kimball**[1], **Don Des Jarlais**[1,3]

**1** Department of Epidemiology, School of Global Public Health, New York University, New York, NY, United States of America, **2** HIV/STI Surveillance Research Center, and WHO Collaborating Center for HIV Surveillance, Institute for Futures Studies in Health, Kerman University of Medical Sciences, Kerman, Iran, **3** Center for Drug Use and HIV/HCV Research, New York, NY, United States of America

* mehrdad.khezri@nyu.edu

**Data Availability Statement:** The data supporting the findings of this study contains Personal Health Information (PHI) which is protected under the Health Insurance Portability and Accountability Act

## Abstract

### Background

Persons who inject drugs (PWID) may be unengaged with healthcare services and face an elevated risk of severe morbidity and mortality associated with COVID-19 due to chronic diseases and structural inequities. However, data on COVID-19 vaccine uptake, particularly booster vaccination, among PWID are limited. We examined COVID-19 vaccine uptake and factors associated with booster vaccination among PWID in New York City (NYC).

### Methods

We recruited PWID using respondent-driven sampling from October 2021 to November 2023 in a survey that included HIV and SARS-CoV-2 antibodies testing. The questionnaire included demographics, COVID-19 vaccination and attitudes, and drug use behaviors.

### Results

Of 436 PWID, 80% received at least one COVID-19 vaccine dose. Among individuals who received at least one COVID-19 vaccine dose, 95% were fully vaccinated. After excluding participants recruited before booster authorization for general adults started in NYC, and those who had never received an initial vaccination, 41% reported having received a COVID-19 booster vaccine dose. COVID-19 booster vaccination was significantly associated with having a high school diploma or GED (adjusted odds ratio (aOR) 1.93; 95% confidence interval (CI) 1.09, 3.48), ever received the hepatitis A/B vaccine (aOR 2.23; 95% CI 1.27, 3.96), main drug use other than heroin/speedball, fentanyl and stimulants (aOR 14.4; 95% CI 2.32, 280), number of non-fatal overdoses (aOR 0.35; 95% CI 0.16, 0.70), and mean vaccination attitude score (aOR 0.94; 95% CI 0.89, 0.98).

### Conclusions

We found a suboptimal level of COVID-19 booster vaccination among PWID, which was consistent with the rates observed in the general population in NYC and the U.S.

(HIPPA), such as vaccination status and whether the individual suffers from any of the "underlying conditions" that would be likely to make a COVID-19 infection more severe. Access to the data can be provided through an approved Data Use Agreement between our institution (New York University) and the institution with which the user is affiliated. Persons wanting to access the data should communicate with the NYU IRB (email contact: ask.humansubjects@nyu.edu) to initiate a Data Use Agreement.

**Funding:** Funding for this study was provided by US NIH/NIDA, Grant 5R01DA003574-39. the funder had no role in study design, data collection and analysis, decision to publish, or preparation of the manuscript.

**Competing interests:** No conflict of interest to report or declare.

Community-based interventions are needed to improve COVID-19 booster vaccination access and uptake among PWID. Attitudes towards vaccination were significant predictors of both primary and booster vaccination uptake. Outreach efforts focusing on improving attitudes towards vaccination and educational programs are essential for reducing hesitancy and increasing booster vaccination uptake among PWID.

## Introduction

Persons who inject drugs (PWID) often have limited engagement with healthcare services, attributed to structural adversities, including the absence of health insurance, transportation challenges, unstable housing, incarceration, stigma, and medical mistrust [1–3]. Moreover, PWID face an increased susceptibility to SARS-CoV-2 infection [4] and are at heightened risk of severe complications from COVID-19 because of underlying medical conditions, such as HIV, viral hepatitis, chronic cardiovascular, kidney, liver, and respiratory diseases [5–7]. COVID-19 vaccines can mitigate these complications, however low uptake contributes to higher morbidity and mortality [8].

COVID-19 vaccines became available to the adult United States (US) population in early 2021 [9]. After the authorization of COVID-19 vaccines, New York City (NYC) implemented various initiatives to promote vaccination. These efforts included an expansive public service information campaign and targeted initiatives aimed at reaching individuals with a heightened risk of COVID-19 disease [10,11]. In September 2021, the CDC initially suggested booster shots for individuals who were older, immunosuppressed, or had underlying medical conditions; however, in November 2021, the CDC expanded this to all individuals 18 years and older [12]. While the initial vaccine series reduces COVID-19 transmission, prevents illness, and lowers mortality, receiving a booster is essential, as it further enhances immunity and targets different variants [13]. Understanding the uptake of COVID-19 booster vaccination among marginalized populations, including PWID, is important to enhance and tailor harm reduction services and healthcare provision and is helpful for future vaccination campaigns for this population.

A growing body of research has documented COVID-19 vaccination uptake and its associated factors among PWID in different settings. For example, 38% of PWID in San Diego [2], 68% in Baltimore [14], 49% in Australia [15], and 40% in Tijuana Mexico [16] reported having at least one COVID-19 vaccine dose. We also previously estimated that 81% of PWID in NYC received at least one COVID-19 vaccine, and 76% were fully vaccinated [17].However, despite the high rate of comorbidities and the increased risk of severe illness and mortality from COVID-19 among PWID, as well as the importance of receiving booster vaccination, studies exploring the uptake of COVID-19 booster vaccines and its associated factors in this population remain limited after the widespread availability of booster vaccines in the US. We aimed to examine COVID-19 vaccine uptake and factors associated with booster vaccination among PWID in NYC.

## Methods

### Study design and recruitment

We recruited PWID using an adapted version of respondent-driven sampling (RDS) between October 5, 2021, and November 28, 2023 [18]. Briefly, 14 initial recruits were enlisted from

public parks and areas near syringe service programs (SSPs) and methadone maintenance treatment programs in Manhattan, locations where PWID were known to gather. The selection of seeds aimed to mirror the demographic features of PWID in NYC in terms of age, gender, and race/ethnicity. However, due to a combination of delayed peer referrals through RDS coupons and the suspension of study activities caused by the Omicron surge of COVID-19 in NYC, disruptions occurred in the peer referral process. Diverse methods were implemented, such as staff recruitment of additional seeds, increasing the number of referral coupons from 3 to 6 for existing participants, and accommodating individuals who had lost their referral coupons to enhance recruitment.

The criteria for eligibility in the study comprised of individuals who were at least 18 years old, had reported injecting substances, such as heroin, fentanyl, cocaine, crack, or methamphetamine within the last 30 days, speak and comprehend English, were able to provide informed consent, and had intentions of residing in the NYC-metro area for the upcoming 6 months. Those PWID meeting these eligibility criteria were enrolled in a 6-month serial cohort study, involving two in-person appointments: at baseline and at the conclusion of the six-month period. Participants were compensated with $30 upon completion of the baseline interview, $50 for participating in the 6-month follow-up interview, and $10 for each successful peer referral, up to a maximum of six referrals.

## Data collection

During each visit, trained interviewers conducted individualized, computer-assisted structured interviews, lasting around 30 minutes. Interviewers utilized the Questionnaire Development System (QDS) software from Nova Research Company in Bethesda, MD, USA to create and administer these interviews. We collected data on demographics, drug use behaviors, overdose experiences, substance use treatment history, COVID-19 infection and vaccination, self-reported HIV and HCV status, and treatment history for these infections.

Additionally, participants underwent an assessment at each visit that included a blood draw for testing HIV, HCV, and SARS-CoV-2 antibodies. HIV, HCV, and SARS-CoV-2 antibody testing was conducted by BioReference Laboratories. HIV testing involved a 4th generation HIV enzyme-linked immunosorbent assay from Siemens (Munich, Germany), and a Geenius assay from Bio-Rad (Hercules, California, USA) with PCR confirmation specifically for HIV-1. HCV antibody testing employed a Siemens chemiluminescence assay (Munich, Germany), while SARS-CoV-2 testing utilized a Roche Elecsys chemiluminescence assay (Roche; Geneva, Switzerland).

## Measures

The primary outcomes for this study were COVID-19 vaccine uptake. Participants were asked if they had received at least one COVID-19 vaccine dose, received both shots of the COVID-19 vaccine, and if they had received a COVID-19 booster vaccine dose.

Covariates included age, gender (men or women), race or ethnicity (Non-Hispanic White, Non-Hispanic Black, Hispanic, or Mixed/Other race), having a high school diploma or GED (yes or no), and the main source of income in the last 6 months (regular employment, government benefits, irregular employment, or friend/relative's income, or possibly illegal). Additional factors taken into account were housing status in the last 6 months (stably housed, housed with friends/relatives, or unstable/homeless), experiences of food insecurity in the last 6 months (yes or no), and whether participants had medical insurance (yes or no). Further covariates about participants' health status and behaviors were assessed, including the results of the COVID-19 antibody lab test (positive, negative, or unknown), mean vaccination attitude

score with standard deviation (SD), history of receiving the hepatitis A/B vaccine (yes, no, or I don't know), Kessler Psychological Distress (serious or moderate/minor), presence of substance use disorder (severe or mild/moderate), ever received psychiatric diagnosis (yes or no), baseline HIV test result (negative, positive, or not successfully collected due to collapsed vein), the number of previous overdoses (more than 3 times, 2–3 times, or 0/1), current main drug use (heroin/speedball, fentanyl, cocaine/crack/methamphetamines, or other), and receiving methadone or buprenorphine maintenance treatment (current, never, or previous).

The COVID-19 vaccination attitude scale, designed to assess attitudes, beliefs, and knowledge related to COVID-19, was formed using 11 items. Each item was rated on a scale of 1 to 4 (strongly agree, agree, disagree, and strongly disagree), with specific items subjected to reverse coding. Lower scores on the scale indicated positive attitudes toward vaccination or alignment with evidence-based public health strategies for addressing the COVID-19 pandemic, commonly referred to as pro-vaccine. The scale demonstrated a high level of reliability, with a Cronbach's α value of 0.81 [17,19].

## Statistical analysis

Descriptive statistics, including absolute and relative frequencies of the main study outcomes and other variables were calculated and reported. Bivariate and multivariate logistic regression models were utilized to identify correlates of COVID-19 vaccination and booster vaccination. The inclusion of individual and environmental covariates in the analysis was based on known associations derived from the literature and the results of the explanatory models. The study reports both unadjusted odds ratios (OR) and adjusted odds ratios (aOR) with corresponding 95% confidence intervals (CIs). Variables demonstrating a P value < 0.2 in the bivariate analysis were incorporated into a comprehensive multivariate logistic regression model. The final model was determined using a backward elimination strategy, with statistical significance established at a P value < 0.05. As the COVID-19 vaccination attitude scale appears to be a significant factor associated with vaccinations, we conducted additional analyses using chi-square tests to examine individual scale items as predictors of both COVID-19 vaccination and booster vaccination. All analyses were conducted using R [20].

## Ethical considerations

Prior to the baseline interview appointment, all participants provided written informed consent after being fully informed about the study's objectives, procedures, and the confidential nature of data collection. The informed consent process emphasized the voluntary nature of participation and the right to withdraw from the study at any time. Participation was confidential and all data collected were de-identified to prevent the identification of individuals. The study was approved by the New York University Institutional Review Board.

## Results

### Participant's characteristics

Of the 436 PWID, the mean age was 48.7 (SD = 10.3) years. Most participants were men (74.0%; n = 322), had a high school diploma or GED (69.0%; n = 300), and received government benefits as the main source of income in the last 6 months (69.0%; n = 301). Overall, 31.7% (n = 138) of participants were non-Hispanic Black, 30.0% (n = 131) were non-Hispanic White, and 30.0% (n = 131) were Hispanic. The majority of PWID reported experiencing unstable housing or homelessness (45.0%; n = 196) and food insecurity (66.5%; n = 290) in the last 6 months. A total of 417 (95.9%) PWID reported having medical insurance, 49.8%

(n = 217) ever received the hepatitis A/B vaccine, and the mean (SD) vaccination attitude score was 24.9 (6.2) (**Table 1**).

### COVID-19 vaccine uptake

Out of 436 PWID, 80.5% (n = 351) received at least one COVID-19 vaccine dose, while 19.5% (n = 85) remained unvaccinated (**Table 1**). Among individuals who received at least one COVID-19 vaccine dose, 95.4% (n = 335) had received two doses and were fully vaccinated. After excluding participants recruited before November 2021 (when booster authorization for the general adult population started in NYC), and those who had never received an initial vaccination, 41.0% (n = 136) of the 332 eligible for a booster reported having received a COVID-19 booster vaccine dose (**Table 2**). They represent 81.2% of total 409 participants recruited during the period in which participants were eligible for receiving primary vaccinations and boosters.

### Factors associated with COVID-19 vaccination

Not receiving a COVID-19 vaccine was significantly higher among PWID who had not received the hepatitis A/B vaccine in their lifetime (26.4% vs. 15.7%, P < 0.001). Additionally, the mean vaccination attitude score (SD) was significantly higher among unvaccinated PWID (30.4 (5.9) vs. 23.9 (5.7), P < 0.001) (**Table 1**).

Among participants recruited after November 2021 and who had received an initial vaccination, PWID who received a COVID-19 booster dose compared to those who did not were older (mean age (SD): 50.6 (9.5) vs. 47.6 (9.9), P = 0.006), and had a more positive (lower) mean vaccination attitude score (22.6 (4.6) vs. 24.8 (6.1), P < 0.001). Moreover, COVID-19 booster vaccination was significantly higher among PWID who had a high school diploma or GED (44.3% vs. 33.0%, P = 0.054), possessed medical insurance (42.2% vs. 8.3%, P = 0.047), had ever received the hepatitis A/B vaccine (48.6% vs. 30.6%, P = 0.002), had mild/moderate substance use disorder compared to severe (61.5% vs. 39.2%, P = 0.030), tested positive for HIV compared to those tested negative (71.4% vs. 38.7%, P = 0.006), and had a lower number of non-fatal drug overdoses (42.6% vs. 25.7% for more than three times, P = 0.015) (**Table 2**).

Multivariable analysis showed that COVID-19 booster vaccination uptake was significantly and positively associated with having a high school diploma or GED (aOR 1.93; 95% CI 1.09, 3.48), ever receiving the hepatitis A/B vaccine (aOR 2.23; 95% CI 1.27, 3.96), and using other drugs as the main drug rather than heroin/speedball, fentanyl, or stimulants (aOR 14.4; 95% CI 2.32, 280). COVID-19 booster vaccination uptake was also significantly and negatively associated with mean vaccination attitude score (aOR 0.94; 95% CI 0.89, 0.98), number of non-fatal overdoses (aOR 0.35; 95% CI 0.16, 0.70) (**Table 3**).

**Tables 4** and **5** present the analyses of individual attitude items regarding attitudes towards vaccination among those who have received at least one COVID-19 vaccine dose and booster vaccination. Considering both analyses, the most significant association was observed with attitudes toward the safety of the COVID-19 vaccine and its importance for health. PWID who reported believing that the COVID-19 vaccine is unsafe reported lower uptake of both primary and booster vaccines (P < 0.001), whereas those who indicated that getting vaccinated is important for their health reported higher uptake of both primary and booster vaccines (P < 0.001).

### SARS-CoV-2 antibody testing

Among the total sample, 87.6% (n = 382) tested positive in antibody testing for SARS-CoV-2. SARS-CoV-2 antibody testing indicated that 92.0% were positive among those who received at

**Table 1. COVID-19 vaccination by sociodemographic characteristics, and structural level and individual level determinants among persons who inject drugs in New York City, 2021–2023.**

| Variable | Total N | COVID-19 vaccine uptake | | | |
|---|---|---|---|---|---|
| | | Received ≥ 1 vaccine dose n (%) | Unvaccinated n (%) | Odds ratio (95% CI) | P value |
| **Overall** | 436 | 351 (80.5) | 85 (19.5) | - | - |
| **Mean age (SD)** | 48.7 (10.3) | 47.6 (11.7) | 48.9 (9.9) | 0.98 (0.96, 1.00) | 0.253 |
| **Gender** | | | | | |
| Men | 322 (74.0) | 263 (81.7) | 59 (18.3) | 1 | |
| Women | 113 (26.0) | 87 (77.0) | 26 (23.0) | 1.33 (0.79, 2.22) | 0.281 |
| **Race or ethnicity** | | | | | |
| Non-Hispanic White | 131 (30.0) | 107 (81.7) | 24 (18.3) | 1 | |
| Non-Hispanic Black | 138 (31.7) | 107 (77.5) | 31 (22.5) | 1.30 (0.71, 2.36) | 0.400 |
| Hispanic | 131 (30.0) | 110 (84.0) | 21 (16.0) | 0.85 (0.44, 1.61) | 0.623 |
| Mixed/Other race | 36 (8.3) | 27 (75.0) | 9 (25.0) | 1.48 (0.59, 3.48) | 0.375 |
| **High school diploma or GED** | | | | | |
| Yes | 300 (69.0) | 243 (81.0) | 57 (19.0) | 1 | |
| No | 135 (31.0) | 107 (79.3) | 28 (20.7) | 1.12 (0.67, 1.84) | 0.672 |
| **Main source of income, last 6 months** | | | | | |
| Regular employment | 36 (8.3) | 31 (86.1) | 5 (13.9) | 1 | |
| Government benefits | 301 (69.0) | 244 (81.1) | 57 (18.9) | 1.45 (0.58, 4.39) | 0.462 |
| Irregular employment or friend/relative's income | 57 (13.1) | 46 (80.7) | 11 (19.3) | 1.48 (0.49, 5.09) | 0.503 |
| Possibly illegal | 42 (9.6) | 30 (71.4) | 12 (28.6) | 2.48 (0.81, 8.58) | 0.124 |
| **Housing status, last 6 months** | | | | | |
| Stably housed | 153 (35.1) | 126 (82.4) | 27 (17.6) | 1 | |
| Housed with friends/relatives | 87 (19.9) | 63 (72.4) | 24 (27.6) | 1.78 (0.95, 3.33) | 0.072 |
| Unstable/homeless | 196 (45.0) | 162 (82.7) | 34 (17.3) | 0.98 (0.56, 1.72) | 0.942 |
| **Experienced food insecurity, last 6 months** | | | | | |
| Yes | 290 (66.5) | 234 (80.7) | 56 (19.3) | 1 | |
| No | 146 (33.5) | 117 (80.1) | 29 (19.9) | 1.04 (0.62, 1.70) | 0.891 |
| **Having medical insurance** | | | | | |
| Yes | 417 (95.9) | 337 (80.8) | 80 (19.2) | 1 | |
| No | 18 (4.1) | 13 (72.2) | 5 (27.8) | 1.62 (0.51, 4.43) | 0.372 |
| **Mean vaccination attitude score (SD)** | 24.9 (6.2) | 23.9 (5.7) | 30.4 (5.9) | 1.19 (1.13, 1.26) | < 0.001 |
| **Ever received the Hepatitis A/B vaccine** | | | | | |
| Yes | 217 (49.8) | 183 (84.3) | 34 (15.7) | 1 | |
| No | 178 (40.8) | 131 (73.6) | 47 (26.4) | 1.93 (1.18, 3.19) | < 0.001 |
| I don't Know | 41 (9.4) | 37 (90.2) | 4 (9.8) | 0.58 (0.17, 1.57) | 0.332 |
| **Kessler Psychological distress** | | | | | |
| Moderate/Minor | 277 (63.5) | 226 (81.6) | 51 (18.4) | 1 | |
| Serious | 159 (36.5) | 125 (78.6) | 34 (21.4) | 1.21 (0.74, 1.95) | 0.451 |
| **Substance use disorder** | | | | | |
| Mild/Moderate | 36 (8.3) | 30 (83.3) | 6 (16.7) | 1 | |
| Severe | 400 (91.7) | 321 (80.2) | 79 (19.8) | 1.23 (0.53, 3.37) | 0.655 |
| **Psychiatric diagnosis, ever** | | | | | |
| No | 180 (41.3) | 145 (80.6) | 35 (19.4) | 1 | |
| Yes | 256 (58.7) | 206 (80.5) | 50 (19.5) | 1.01 (0.62, 1.64) | 0.982 |
| **HIV test result, baseline** | | | | | |

*(Continued)*

**Table 1.** (Continued)

| Variable | | COVID-19 vaccine uptake | | | |
|---|---|---|---|---|---|
| | Total N | Received ≥ 1 vaccine dose n (%) | Unvaccinated n (%) | Odds ratio (95% CI) | P value |
| Negative | 370 (84.9) | 298 (80.5) | 72 (19.5) | 1 | |
| Positive | 25 (5.7) | 22 (88.0) | 3 (12.0) | 0.56 (0.13, 1.69) | 0.363 |
| Not successfully collected due to collapsed vein | 41 (9.4) | 31 (75.6) | 10 (24.4) | 1.34 (0.60, 2.76) | 0.455 |
| **Number of previous overdoses** | | | | | |
| 0/1 | 237 (54.4) | 192 (81.0) | 45 (19.0) | 1 | |
| 2–3 times | 107 (24.5) | 88 (82.2) | 19 (17.8) | 0.92 (0.50, 1.65) | 0.786 |
| More than 3 times | 92 (21.1) | 71 (77.2) | 21 (22.8) | 1.26 (0.69, 2.24) | 0.436 |
| **Main drug use, current** | | | | | |
| Heroin/Speedball | 342 (78.4) | 271 (79.2) | 71 (20.8) | 1 | |
| Fentanyl | 33 (7.6) | 28 (84.8) | 5 (15.2) | 0.68 (0.23, 1.69) | 0.446 |
| Cocaine/crack/Methamphetamines | 47 (10.8) | 41 (87.2) | 6 (12.8) | 0.56 (0.21, 1.28) | 0.203 |
| Other [a] | 14 (3.2) | 11 (78.6) | 3 (21.4) | 1.04 (0.23, 3.44) | 0.952 |
| **Receiving methadone or buprenorphine maintenance treatment** | | | | | |
| Current | 235 (54.0) | 194 (82.6) | 41 (17.4) | 1 | |
| Never | 84 (19.3) | 64 (76.2) | 20 (23.8) | 1.48 (0.80, 2.68) | 0.205 |
| Previous | 116 (26.7) | 92 (79.3) | 24 (20.7) | 1.23 (0.70, 2.15) | 0.462 |

[a] Other drugs included benzodiazepines, opiate analgesics, and street methadone.

least one COVID-19 vaccine dose, and 69.4% tested positive among those who were unvaccinated (P < 0.001).

## Discussion

Our study showed that over two-thirds of PWID in NYC had received at least one COVID-19 vaccine dose, of which over one-third had received a booster dose. COVID-19 booster vaccination was significantly associated with having a high school diploma or GED, more positive overall views about vaccination, ever receiving the hepatitis A/B vaccine, lower number of non-fatal overdoses, and main drug use other than heroin/speedball, fentanyl and stimulants. Nearly nine in ten PWID tested positive for SARS-CoV-2 antibodies. Among individuals who received at least one dose of the COVID-19 vaccine, 92% were found to have positive results in SARS-CoV-2 antibody testing, while among those unvaccinated, 69% tested positive indicating natural infection.

The COVID-19 vaccine coverage among PWID in NYC was found to be comparable to the vaccination rates in the general population of NYC and the US. According to the Centers for Disease Control and Prevention, the primary series vaccination rates for all age groups were 80% for the NYC metro area and 69% for the US. The bivalent booster rate was 16% for the NYC metro area and 17% for the US until December 7, 2023 [21]. The higher vaccination rate in NYC compared to the US supports the notion that the city implemented significant initiatives to promote vaccination within the general population, with particular attention directed towards individuals deemed more susceptible to severe cases of COVID-19 [10,11]. Considering the difficulties PWID face in getting COVID-19 shots due to substance use disorders, unemployment, and unstable housing, our estimates for vaccination and booster shots among PWID support the effectiveness of targeted initiatives in NYC to reach individuals at higher risk.

**Table 2. COVID-19 booster vaccination among persons who inject drugs who recruited after November 2021 and received an initial vaccination in New York City, 2021–2023.**

| Variable | Total N | COVID-19 booster vaccination uptake | | | |
|---|---|---|---|---|---|
| | | No n (%) | Yes n (%) | Odds ratio (95% CI) | P value |
| **Overall [a]** | 332 | 196 (59.0) | 136 (41.0) | - | - |
| **Mean age (SD)** | 48.8 (9.9) | 47.6 (9.9) | 50.6 (9.5) | 1.03 (1.01, 1.06) | 0.006 |
| **Gender** | | | | | |
| Men | 248 (75.0) | 142 (57.3) | 106 (42.7) | 1 | |
| Women | 83 (25.0) | 54 (65.1) | 29 (34.9) | 0.72 (0.43, 1.20) | 0.212 |
| **Race or ethnicity** | | | | | |
| Non-Hispanic White | 102 (30.7) | 66 (64.7) | 36 (35.3) | 1 | |
| Non-Hispanic Black | 101 (30.4) | 59 (58.4) | 42 (41.6) | 1.31 (0.74, 2.31) | 0.357 |
| Hispanic | 106 (31.9) | 59(55.7) | 47 (44.3) | 1.46 (0.84, 2.56) | 0.184 |
| Mixed/Other race | 23 (6.9) | 12 (52.2) | 11 (47.8) | 1.68 (0.67, 4.21) | 0.265 |
| **High school diploma or GED** | | | | | |
| No | 103 (31.2) | 69 (67.0) | 34 (33.0) | 1 | |
| Yes | 228 (68.8) | 127 (55.7) | 101 (44.3) | 1.61 (1.00, 2.65) | 0.054 |
| **Main source of income in last 6 months** | | | | | |
| Regular employment | 29 (8.7) | 20 (69.0) | 9 (31.0) | 1 | |
| Government benefits | 231 (69.6) | 133 (57.6) | 98 (42.4) | 1.64 (0.73, 3.92) | 0.244 |
| Irregular employment or friend/relative's income | 45 (13.6) | 26 (57.8) | 19 (42.2) | 1.62 (0.62, 4.48) | 0.334 |
| Possibly illegal | 27 (8.1) | 17 (63.0) | 10 (37.0) | 1.31 (0.43, 4.03) | 0.636 |
| **Housing status, last 6 months** | | | | | |
| Stably housed | 118 (35.5) | 67 (56.8) | 51 (43.2) | 1 | |
| Housed with friends/relatives | 56 (16.9) | 37 (66.1) | 19 (33.9) | 0.67 (0.34, 1.30) | 0.244 |
| Unstable/homeless | 158 (47.6) | 92 (58.2) | 66 (41.8) | 0.94 (0.58, 1.53) | 0.810 |
| **Food insecurity, last 6 months** | | | | | |
| Yes | 226 (68.1) | 141 (62.4) | 85 (37.6) | 1 | |
| No | 106 (31.9) | 55 (51.9) | 51 (48.1) | 1.54 (0.96, 2.46) | 0.070 |
| **Having medical insurance** | | | | | |
| No | 12 (3.6) | 11 (91.7) | 1 (8.3) | 1 | |
| Yes | 320 (96.4) | 185 (57.8) | 135 (42.2) | 8.03 (1.53, 148) | 0.047 |
| **Mean vaccination attitude score (SD)** | 23.9 (5.6) | 24.8 (6.1) | 22.6 (4.6) | 0.93 (0.89, 0.97) | <0.001 |
| **Ever received the Hepatitis A/B vaccine** | | | | | |
| No | 121 (36.5) | 84 (69.4) | 37 (30.6) | 1 | |
| Yes | 177 (53.3) | 91 (51.4) | 86 (48.6) | 2.15 (1.33, 3.51) | 0.002 |
| I don't Know | 34 (10.2) | 21 (61.8) | 13 (38.2) | 1.41 (0.63, 3.08) | 0.400 |
| **Kessler Psychological distress** | | | | | |
| Serious | 117 (35.2) | 73 (62.4) | 44 (37.6) | 1 | |
| Moderate/Minor | 215 (64.8) | 123 (57.2) | 92 (42.8) | 1.24 (0.78, 1.98) | 0.359 |
| **Substance use disorder** | | | | | |
| Severe | 306 (92.2) | 186 (60.8) | 120 (39.2) | 1 | |
| Mild/Moderate | 26 (7.8) | 10 (38.5) | 16 (61.5) | 2.48 (1.10, 5.83) | 0.030 |
| **Psychiatric diagnosis, ever** | | | | | |
| No | 135 (40.7) | 76 (56.3) | 59 (43.7) | 1 | |
| Yes | 197 (59.3) | 120 (60.9) | 77 (39.1) | 0.83 (0.53, 1.29) | 0.401 |
| **HIV test result, baseline** | | | | | |
| Negative | 282 (84.9) | 173 (61.3) | 109 (38.7) | 1 | |
| Positive | 21 (6.3) | 6 (28.6) | 15 (71.4) | 3.97 (1.56, 11.4) | 0.006 |

*(Continued)*

**Table 2.** (Continued)

| Variable | | COVID-19 booster vaccination uptake | | | |
| --- | --- | --- | --- | --- | --- |
| | Total N | No n (%) | Yes n (%) | Odds ratio (95% CI) | P value |
| Not successfully collected due to collapsed vein | 29 (8.8) | 17 (58.6) | 12 (41.4) | 1.12 (0.50, 2.42) | 0.774 |
| **Number of previous overdoses** | | | | | |
| 0/1 | 176 (53.0) | 101 (57.4) | 75 (42.6) | 1 | |
| 2–3 times | 86 (25.9) | 43 (50.0) | 43 (50.0) | 1.35 (0.80, 2.26) | 0.260 |
| More than 3 times | 70 (21.1) | 52 (74.3) | 18 (25.7) | 0.47 (0.25, 0.85) | 0.015 |
| **Main drug use, current** | | | | | |
| Heroin/Speedball | 255 (76.8) | 154 (60.4) | 101 (39.6) | 1 | |
| Fentanyl | 27 (8.1) | 20 (74.1) | 7 (25.9) | 0.53 (0.20, 1.25) | 0.170 |
| Cocaine/crack/Methamphetamines | 39 (11.8) | 18 (46.2) | 21 (53.8) | 1.78 (0.90, 3.53) | 0.096 |
| Other | 11 (3.3) | 4 (36.4) | 7 (63.6) | 2.67 (0.79, 10.4) | 0.125 |
| **Receiving methadone or buprenorphine maintenance treatment** | | | | | |
| Current | 182 (55.0) | 114 (62.6) | 68 (37.4) | 1 | |
| Never | 63 (19.0) | 34 (54.0) | 29 (46.0) | 1.43 (0.80, 2.55) | 0.226 |
| Previous | 86 (26.0) | 48 (55.8) | 38 (44.2) | 1.33 (0.79, 2.23) | 0.287 |

[a] Overall, after excluding participants who were recruited before November 2021 and those who had never received an initial vaccination.

Moreover, our study reported a higher COVID-19 vaccine coverage compared to studies conducted among PWID in Baltimore [14], Oregon [3], San Diego [2], Tijuana, Mexico [16], and Australia [15]. Differences in study methods, data collection dates (most studies collected

**Table 3.** Multivariable analysis of associations of COVID-19 booster vaccination among persons who inject drugs in New York City, 2021–2023.

| Variable | Adjusted odds ratio (95% CI) [a] | P value |
| --- | --- | --- |
| **High school diploma or GED** | | |
| No | 1 | |
| Yes | 1.93 (1.09, 3.48) | 0.026 |
| **Mean vaccination attitude score (SD)** | 0.94 (0.89, 0.98) | 0.007 |
| **Ever received the Hepatitis A/B vaccine** | | |
| No | 1 | |
| Yes | 2.23 (1.27, 3.96) | 0.005 |
| I don't Know | 1.37 (0.56, 3.30) | 0.482 |
| **Number of previous overdoses** | | |
| 0/1 | 1 | |
| 2–3 times | 1.34 (0.74, 2.44) | 0.339 |
| More than 3 times | 0.35 (0.16, 0.70) | 0.005 |
| **Main drug use, current** | | |
| Heroin/Speedball | 1 | |
| Fentanyl | 0.50 (0.18, 1.30) | 0.173 |
| Cocaine/crack/Methamphetamines | 1.36 (0.62, 2.98) | 0.441 |
| Other [b] | 14.4 (2.32, 280) | 0.016 |

[a] Using multivariable logistic regression, variables with a P value < 0.2 in the bivariable analysis were entered into the multivariable analysis. The final model was selected through a backward elimination approach with significance was set at P value < 0.05.

[b] Other drugs included benzodiazepines, opiate analgesics, and street methadone.

**Table 4. Receiving at least one COVID-19 vaccine dose by Anti-Vaccine Attitudes Scale among persons who inject drugs in New York City, 2021–2023.**

| Variable | Received ≥ 1 vaccine dose | | |
|---|---|---|---|
| | No n (%) | Yes n (%) | P value |
| **Item–Positively scored [a]** | | | |
| **I do not like vaccines in general** | | | |
| Agree | 55 (29.6) | 131 (70.4) | < 0.001 |
| Disagree | 28 (11.4) | 217 (88.6) | |
| **I do not trust pharmaceutical companies in general** | | | |
| Agree | 58 (24.7) | 177 (75.3) | 0.002 |
| Disagree | 25 (12.8) | 170 (87.2) | |
| **I believe that the dangers of COVID have been greatly exaggerated** | | | |
| Agree | 48 (26.4) | 134 (73.6) | 0.002 |
| Disagree | 36 (14.6) | 211 (85.4) | |
| **Even if I got infected or re-infected, I do not think I would get seriously ill from COVID-19** | | | |
| Agree | 38 (24.2) | 119 (75.8) | 0.030 |
| Disagree | 41 (15.6) | 221 (84.4) | |
| **I think the COVID-19 vaccine is unsafe** | | | |
| Agree | 56 (42.4) | 76 (57.6) | < 0.001 |
| Disagree | 25 (8.6) | 266 (91.4) | |
| **I know of family/friends who have gotten the COVID-19 vaccine** | | | |
| Agree | 76 (18.8) | 328 (81.2) | 0.540 |
| Disagree | 7 (23.3) | 23 (76.7) | |
| **Item–Reverse Scored I [a]** | | | |
| **The COVID-19 vaccine is very good at preventing severe COVID-19 disease** | | | |
| Agree | 32 (11.0) | 260 (89.0) | < 0.001 |
| Disagree | 39 (33.1) | 79 (66.9) | |
| **I trust information I receive from government health agencies about the COVID-19 vaccine** | | | |
| Agree | 26 (12.4) | 183 (87.6) | < 0.001 |
| Disagree | 57 (25.7) | 165 (74.3) | |
| **Overall vaccines are safe** | | | |
| Agree | 37 (11.2) | 293 (88.8) | < 0.001 |
| Disagree | 42 (45.7) | 50 (54.3) | |
| **Overall vaccines are effective** | | | |
| Agree | 44 (12.6) | 304 (87.4) | < 0.001 |
| Disagree | 35 (46.1) | 41 (53.9) | |

*(Continued)*

**Table 4.** (Continued)

| Variable | Received ≥ 1 vaccine dose | | |
|---|---|---|---|
| | No n (%) | Yes n (%) | P value |
| **Getting vaccinated is important for my health** | | | |
| Agree | 37 (10.6) | 313 (89.4) | < 0.001 |
| Disagree | 43 (55.1) | 35 (44.9) | |

ᵃ For generating the continuous variable in Tables 1 and 2, each scale scored from 1 (strongly agree), 2 (agree) 3 (disagree) 4 (strongly disagree) for positively scored items. From 1 (strongly disagree), 2 (disagree) 3 (agree) 4 (strongly disagree) for reverse scored items. Lower scores indicate more positive attitudes, higher scores more negative attitudes toward COVID-19 vaccination.

data in the early phase of vaccination), and settings contribute to the variations observed in our study regarding the receipt of at least one COVID-19 vaccine dose. However, no previous study has specifically focused on booster vaccination among PWID who received the initial dose but did not continue with their immunizations. Our data suggest that less than half of those who had received an initial vaccination reported having received a COVID-19 booster vaccine dose. This suboptimal booster vaccination rate leaves this population at significant risk for SARS-CoV-2 (re)infection, as the COVID-19 pandemic persists. The comparable vaccination rates among PWID in NYC, in comparison to the general population in both NYC and the US, support the idea of extending health services to PWID, who are likely to respond positively to such efforts.

We found that individuals with lower education levels and anti-vaccine attitudes, particularly, towards the safety of the COVID-19 vaccine and its importance for health, were less likely to receive the COVID-19 booster vaccination. This could be explained by the fact that higher levels of education are often associated with a better understanding of the importance of vaccination, as consistent with findings in the general population in the US [22]. Attitudes towards vaccination can also be influenced by cultural beliefs, previous experiences with healthcare, trust in healthcare providers, and COVID-19 disinformation [2,23]. These factors suggest that anti-vaccine attitudes play a role in influencing COVID-19 vaccine uptake. These findings are consistent with the existing literature on vaccine uptake among PWID and the US general population. For example, PWID in San Diego, holding the belief that COVID-19 vaccines contained a tracking device, reported lower COVID-19 vaccination rates, highlighting the role of disinformation spread through social media [2]. Another study conducted among PWID in San Diego and Tijuana, Mexico, also found a significant correlation between COVID-19-related disinformation and vaccine hesitancy in this population [23]. In a national study of US adults, conspiracy beliefs related to COVID-19 were associated with resistance to uptake of preventive behaviors and vaccination [24]. These findings underscore the necessity for targeted interventions to enhance vaccine trust and uptake in this marginalized population. Research indicates that attitudes can evolve, and interventions aimed at enhancing health literacy and debunking vaccine-related misconceptions could improve COVID-19 vaccine acceptance among PWID initially expressing hesitation [2,3]. Additionally, community-based interventions involving peers and outreach efforts should prioritize debunking conspiracy theories and promoting health literacy among PWID [25].

Our analysis also demonstrated that PWID who ever received the hepatitis A/B vaccine were more likely to have received a COVID-19 booster vaccination. This finding is in agreement with previous research. Studies in Baltimore and San Diego reported that PWID who

**Table 5. COVID-19 booster vaccination by anti-vaccine attitudes scale among persons who inject drugs in New York City, 2021–2023.**

| Variable | COVID-19 booster vaccination | | |
|---|---|---|---|
| | No n (%) | Yes n (%) | P value |
| **Item–Positively scored [a]** | | | |
| **I do not like vaccines in general** | | | |
| Agree | 85 (68.0) | 40 (32.0) | 0.007 |
| Disagree | 108 (52.9) | 96 (47.1) | |
| **I do not trust pharmaceutical companies in general** | | | |
| Agree | 97 (59.1) | 67 (40.9%) | 0.99 |
| Disagree | 97 (59.1) | 67 (40.9) | |
| **I believe that the dangers of COVID have been greatly exaggerated** | | | |
| Agree | 81 (65.3) | 43 (34.7) | 0.053 |
| Disagree | 110 (54.5) | 92 (45.5) | |
| **Even if I got infected or re-infected, I do not think I would get seriously ill from COVID-19** | | | |
| Agree | 57 (52.8) | 51 (47.2) | 0.106 |
| Disagree | 133 (62.1) | 81 (37.9) | |
| **I think the COVID-19 vaccine is unsafe** | | | |
| Agree | 59 (84.3) | 11 (15.7) | < 0.001 |
| Disagree | 132 (52.0) | 122 (48.0) | |
| **I know of family/friends who have gotten the COVID-19 vaccine** | | | |
| Agree | 181 (58.2) | 130 (41.8) | 0.233 |
| Disagree | 15 (71.4) | 6 (28.6) | |
| **Item–Reverse Scored I [a]** | | | |
| **The COVID-19 vaccine is very good at preventing severe COVID-19 disease** | | | |
| Agree | 139 (56.7) | 106 (43.3) | 0.385 |
| Disagree | 48 (62.3) | 29 (37.7) | |
| **I trust information I receive from government health agencies about the COVID-19 vaccine** | | | |
| Agree | 90 (52.6) | 81 (47.4) | 0.021 |
| Disagree | 103 (65.2) | 55 (34.8) | |
| **Overall vaccines are safe** | | | |
| Agree | 154 (55.6) | 123 (44.4) | 0.012 |
| Disagree | 36 (75.0) | 12 (25.0) | |
| **Overall vaccines are effective** | | | |
| Agree | 165 (57.1) | 124 (42.9) | 0.023 |
| Disagree | 29 (76.3) | 9 (23.7) | |
| **Getting vaccinated is important for my health** | | | |
| Agree | 167 (56.0) | 131 (44.0) | < 0.001 |
| Disagree | 29 (90.6) | 3 (9.4) | |

had received influenza vaccination had higher level of receiving a COVID-19 vaccine dose and lower levels of vaccine hesitancy [2,14]. In the US general population, attitudes towards COVID-19 vaccines are also associated with the overall global acceptance or hesitancy towards vaccination [26]. Overall, these findings suggest the integration of vaccination campaigns for COVID-19, influenza, and Hepatitis A and B in this population to improve vaccine uptake [2,26].

Our antibody testing showed that about nine in ten PWID in our sample tested positive for SARS-CoV-2 antibody. Overall, 92% of those received at least a COVID-19 vaccine does, while 69% of unvaccinated PWID tested positive for SARS-CoV-2 antibody. This suggests a decrease in antibody levels among those who received the vaccine, emphasizing the necessity of booster doses. It also highlights a significant proportion of unvaccinated participants who have antibodies due to natural infection. Among PWID, research on SARS-CoV-2 antibody testing is limited; however, studies in Australia and San Diego reported that previous SARS-CoV-2 testing was an independent predictor of vaccine uptake, suggesting that initiatives aimed at broadening the scope of COVID-19 case identification within this population could influence the coverage of COVID-19 vaccination [2,15].

The allocation of limited vaccination resources underscores the need for prioritizing vaccination initiatives and necessitates a targeted approach. While the reasons for vaccine hesitancy among PWID are complex, key attitudes hindering the uptake of booster vaccinations include a general aversion to vaccination and concerns about the safety of COVID-19 vaccines. A national study among the general population in the US also suggested that the acceptance of booster doses was primarily linked to a strong belief in the necessity of vaccination, trust in the safety of vaccines, and concerns about contracting COVID-19 [13]. This suggests the crucial need for implementing innovative strategies targeted at enhancing confidence in vaccines, particularly COVID-19 vaccines, and improving COVID-19 risk perception among PWID to effectively boost vaccine uptake in this population. Qualitative research among PWID in an NYC SSP found that fears of potential side effects, combined with medical mistrust and doubt regarding the overall value of vaccination, contributed to significant ambivalence among this population [27]. For example, participants reported concerns regarding the safety of the vaccines, and whether their bodies' potential vulnerability due to injection drug use has heightened the risk of potential vaccine side effects. This suggests that community-developed messages through outreach efforts are essential to clarify the importance of vaccination, highlighting the significantly higher risks of COVID-19 in contrast to potential unintended side effects. Providing COVID-19 vaccination services, including booster doses through organizations that currently offer services to PWID, such as SSPs, have been suggested to facilitate easy access to vaccines on-site [27,28].

## Limitations

Our results should be considered in the context of several limitations. First, data collected were cross sectional and prevents the ability to infer causal associations. Second, data on self-report for COVID-19 vaccination and high-risk behaviors were collected using face-to-face interviews, which could be subject to social desirability and under-reporting biases. To help address this issue, we utilized experienced interviewers and ensured they received the required training. Third, although we employed RDS techniques for participants recruitment, our efforts were disrupted by the Omicron surge, and we faced constraints imposed by COVID-19 protocols at the research site. Because of this, we employed an adapted version of RDS, which can be considered convenience sampling. Fourth, we depended on individuals' self-report attitudes towards vaccination, acknowledging that the attitudes towards vaccination may change over

time. Lastly, the results of this study may not be generalizable to other parts of the US and may not be comparable to different time periods within the COVID-19 pandemic. Despite these limitations, self-reported vaccination status was strongly associated with the presence of SARS-CoV-2 antibodies and remained consistent with individuals' attitudes toward vaccination.

## Conclusions

In summary, we found a suboptimal level of COVID-19 booster vaccination among community-recruited PWID in NYC, which was consistent with the rates observed in the general population in both NYC and the US and associated with the level of education, drug-related characteristics, and attitude towards vaccination. Community-based interventions, including outreach efforts and education programs, are needed to improve COVID-19 booster vaccination access and uptake. Since attitudes towards vaccination were significant predictors of primary and booster vaccination uptake, the findings suggest that targeted interventions to prevent disinformation and medical distrust can enhance booster vaccination uptake and reduce vaccination hesitancy among PWID.

## Acknowledgments

We acknowledge all participants for contributing to the study and their time. MK and SK are supported by NYU Doctoral Fellowships.

## Author Contributions

**Conceptualization:** Mehrdad Khezri, Courtney McKnight, Don Des Jarlais.

**Data curation:** Mehrdad Khezri, Courtney McKnight, Sarah Kimball, Don Des Jarlais.

**Formal analysis:** Mehrdad Khezri, Chenziheng Allen Weng, Don Des Jarlais.

**Funding acquisition:** Courtney McKnight, Don Des Jarlais.

**Investigation:** Courtney McKnight, Sarah Kimball, Don Des Jarlais.

**Methodology:** Mehrdad Khezri, Courtney McKnight, Sarah Kimball, Don Des Jarlais.

**Project administration:** Courtney McKnight, Don Des Jarlais.

**Resources:** Courtney McKnight, Don Des Jarlais.

**Software:** Mehrdad Khezri, Chenziheng Allen Weng.

**Supervision:** Courtney McKnight, Don Des Jarlais.

**Validation:** Don Des Jarlais.

**Visualization:** Don Des Jarlais.

**Writing – original draft:** Mehrdad Khezri.

**Writing – review & editing:** Courtney McKnight, Chenziheng Allen Weng, Sarah Kimball, Don Des Jarlais.

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
