## [Decision Letter · Decision Letter 0]

5 Mar 2024

PONE-D-24-02131COVID-19 vaccination uptake and determinants of booster vaccination among persons who inject drugs in New York CityPLOS ONE

Dear Dr. Khezri,

Thank you for submitting your manuscript to PLOS ONE. After careful consideration, we feel that it has merit but does not fully meet PLOS ONE’s publication criteria as it currently stands. Therefore, we invite you to submit a revised version of the manuscript that addresses the points raised during the review process.

 Please review the manuscript in line with the reviewers's expert's review below

We look forward to receiving your revised manuscript.

Kind regards,

Dr Moses Katbi

Academic Editor

PLOS ONE

“Funding for this study was provided by US NIH/NIDA , Grant 5R01DA003574-39.”

“Funding for this study was provided by US NIH/NIDA , Grant 5R01DA003574-39.”

“Funding for this study was provided by US NIH/NIDA , Grant 5R01DA003574-39.”

4. For studies involving third-party data, we encourage authors to share any data specific to their analyses that they can legally distribute. PLOS recognizes, however, that authors may be using third-party data they do not have the rights to share. When third-party data cannot be publicly shared, authors must provide all information necessary for interested researchers to apply to gain access to the data. (https://journals.plos.org/plosone/s/data-availability#loc-acceptable-data-access-restrictions)

a) A description of the data set and the third-party source

b) If applicable, verification of permission to use the data set

c) Confirmation of whether the authors received any special privileges in accessing the data that other researchers would not have

d) All necessary contact information others would need to apply to gain access to the data

Reviewers' comments:

Reviewer's Responses to Questions

**Comments to the Author**

1. Is the manuscript technically sound, and do the data support the conclusions?

Reviewer #1: Yes

2. Has the statistical analysis been performed appropriately and rigorously? 

Reviewer #1: Yes

3. Have the authors made all data underlying the findings in their manuscript fully available?

Reviewer #1: Yes

4. Is the manuscript presented in an intelligible fashion and written in standard English?

Reviewer #1: Yes

5. Review Comments to the Author

Reviewer #1: How did the researchers ensure the reliability and validity of the data collected, especially considering the sensitive nature of the topics discussed?

How did the researchers address potential biases associated with self-reported vaccination status and attitudes towards vaccination? Were any measures taken to validate the accuracy of self-reported data?

Could you provide more information on the statistical methods used to analyze the data, including the criteria for variable selection in the multivariable regression models? How were potential confounding variables accounted for in the analysis, and what sensitivity analyses were conducted to assess the robustness of the results?

What ethical considerations were considered in the conduct of the study, particularly regarding the recruitment of participants from marginalized populations such as PWID? How were issues of informed consent, privacy, and confidentiality addressed throughout the research process?

The study identifies education level, vaccination attitudes, and drug-related characteristics as significant factors associated with booster vaccination uptake among PWID. Could you elaborate on the potential mechanisms underlying these associations and how they could inform targeted intervention strategies?

Based on the findings, what specific recommendations would the authors propose for public health interventions aimed at improving COVID-19 booster vaccination uptake among PWID in NYC? How might these recommendations be implemented in practice, considering the unique challenges faced by this population?

6. PLOS authors have the option to publish the peer review history of their article (what does this mean?). If published, this will include your full peer review and any attached files.

Reviewer #1: **Yes: **Daniela A. Rodrigues

---

## [Author Response · Author response to Decision Letter 0]

4 Apr 2024

March 27, 2024

Academic Editor

PLOS ONE

Dear Dr. Moses Katbi,

I am writing in regard to our manuscript entitled, “COVID-19 vaccination uptake and determinants of booster vaccination among persons who inject drugs in New York City,” which was submitted to PLOS ONE.

We would like to express our gratitude to the reviewer for the valuable time and constructive and comprehensive suggestions. We have carefully considered their suggestions and made the necessary revisions to improve the manuscript. Below, we have outlined the changes made to the manuscript with detailed responses to the editors’ and reviewers’ comments. Moreover, we confirm that the funder had no role in study design, data collection and analysis, decision to publish, or preparation of the manuscript.

We hope that the revised manuscript will be favorably considered for publication in PLOS ONE. 

Best regards,

Mehrdad Khezri, MSc

Response: Thank you for providing the PLOS ONE style templates. We ensured that our manuscript meets all the style requirements, including file naming. 

“Funding for this study was provided by US NIH/NIDA , Grant 5R01DA003574-39.”

Response: Thank you for acknowledging the financial disclosure provided. The funders, US NIH/NIDA (Grant 5R01DA003574-39), had no role in the study design, data collection and analysis, decision to publish, or preparation of the manuscript. We have included this amended Role of Funder statement in our cover letter as requested.

“Funding for this study was provided by US NIH/NIDA , Grant 5R01DA003574-39.”

We note that you have provided funding information that is currently declared in your Funding Statement. However, funding information should not appear in the Acknowledgments section or other areas of your manuscript. We will only publish funding information present in the Funding Statement section of the online submission form. Please remove any funding-related text from the manuscript and let us know how you would like to update your Funding Statement. Currently, your Funding Statement reads as follows:

“Funding for this study was provided by US NIH/NIDA , Grant 5R01DA003574-39.”

Response: Thank you for bringing this to our attention. We apologize for the oversight. We ensure that the funding information is only included in the Funding Statement section of the online submission form, as per your guidelines. We removed the funding information in the Acknowledgments section and included the Role of Funder statement in our cover letter as requested.

4. For studies involving third-party data, we encourage authors to share any data specific to their analyses that they can legally distribute. PLOS recognizes, however, that authors may be using third-party data they do not have the rights to share. When third-party data cannot be publicly shared, authors must provide all information necessary for interested researchers to apply to gain access to the data. (https://journals.plos.org/plosone/s/data-availability#loc-acceptable-data-access-restrictions)

a) A description of the data set and the third-party source

b) If applicable, verification of permission to use the data set

c) Confirmation of whether the authors received any special privileges in accessing the data that other researchers would not have

d) All necessary contact information others would need to apply to gain access to the data

Response: Thank you for the guidance on data availability requirements. Due to the sensitive nature of the data containing Personal Health Information (PHI) protected under HIPAA, including vaccination status and underlying conditions relevant to COVID-19 severity, we are unable to publicly share it. However, we are committed to facilitating access through an approved Data Use Agreement between our institution and interested researchers’ affiliated institutions. We checked the required information for the Data Availability Statement. The statements reads: 

“The data supporting the findings of this study contains Personal Health Information (PHI) which is protected under the Health Insurance Portability and Accountability Act (HIPPA), such as vaccination status and whether the individual suffers from any of the “underlying conditions” that would be likely to make a COVID-19 infection more severe. Access to the data can be provided through an approved Data Use Agreement between our institution (New York University) and the institution with which the user is affiliated. Persons wanting to access the data should communicate with the NYU IRB (email contact: ask.humansubjects@nyu.edu) to initiate a Data Use Agreement.”

Reviewer #1: 

1. How did the researchers ensure the reliability and validity of the data collected, especially considering the sensitive nature of the topics discussed?

Response: We appreciate your valuable inquiry regarding the reliability and validity of the data collected, especially considering the sensitive nature of the topics addressed in our study. To ensure data quality, we implemented robust data collection protocols and utilized validated survey instruments. Additionally, efforts were taken to maintain participant confidentiality and privacy throughout the research process. However, we acknowledge in the limitation section of the paper that despite these efforts, inherent limitations exist in self-reported data and the potential for response bias due to the sensitive nature of the topics. We emphasize the importance of interpreting our findings within this context in the limitation section: 

“Our results should be considered in the context of several limitations. First, data collected were cross sectional and prevents the ability to infer causal associations. Second, data on self-report for COVID-19 vaccination and higher risk behaviors were collected using face-to-face interviews, which could be subject to social desirability and under-reporting biases. To help address this issue, we utilized experienced interviewers and ensured they received the required training.”

2. How did the researchers address potential biases associated with self-reported vaccination status and attitudes towards vaccination? Were any measures taken to validate the accuracy of self-reported data?

Response: To address potential biases associated with self-reported vaccination status and attitudes towards vaccination, several measures were implemented. Firstly, participants were assured of the confidentiality of their responses. Additionally, prior to data collection, interviewers received training to reduce response bias. It is noteworthy that in the history of this study (grant R01 DA 003574), self-reported data on drug use and sexual behavior have consistently correlated with critical biological variables, particularly HIV infection. For instance, recent modeling of HIV transmission using this survey data demonstrated a notable consistency between modeled HIV incidence and prevalence and empirical studies. This underscores the reliability of the self-reported data in capturing essential health outcomes. Furthermore, our findings show a compelling congruence between reported behaviors, such as vaccination uptake and attitudes towards vaccination, and corresponding biological markers. Notably, our data support the validity of self-reported vaccination status, with a high antibody prevalence of 92% among those who reported receiving vaccination. Such alignment between reported behaviors and biological outcomes further reinforces the credibility of our findings.

Reference: 

Des Jarlais D, Bobashev G, Feelemyer J, McKnight C. Modeling HIV transmission among persons who inject drugs (PWID) at the “End of the HIV Epidemic” and during the COVID-19 pandemic. Drug and alcohol dependence. 2022 Sep 1;238:109573.

3. Could you provide more information on the statistical methods used to analyze the data, including the criteria for variable selection in the multivariable regression models? How were potential confounding variables accounted for in the analysis, and what sensitivity analyses were conducted to assess the robustness of the results?

Response: For the analysis of the data, we utilized multivariable logistic regression models. The criteria for variable selection in these models were based on a combination of theoretical relevance, previous literature, and statistical significance. To account for potential confounding variables in the analysis, we included variables with a P value < 0.2 in the bivariable analysis into the multivariable analysis, and the final model was selected through a backward elimination approach with significance set at P value < 0.05. Furthermore, we conducted additional analyses to assess receiving at least one COVID-19 vaccine dose and booster vaccination by each anti-vaccine attitudes scale. These analyses confirmed the consistency and reliability of our results for the association of vaccination status and attitudes towards vaccination. As the associations were very strong (with large effect sizes), sensitivity analyses were not deemed necessary. To address your comment, we elaborated further on our statistical analysis section to provide more details. The revised statistical analysis section reads: 

“Statistical analysis

Descriptive statistics, including absolute and relative frequencies of the main study outcomes and other variables were calculated and reported. Bivariate and multivariate logistic regression models were utilized to identify correlates of COVID-19 vaccination and booster vaccination. The inclusion of individual and environmental covariates in the analysis was based on known associations derived from the literature and the results of the explanatory models. The study reports both unadjusted odds ratios (OR) and adjusted odds ratios (aOR) with corresponding 95% confidence intervals (CIs). Variables demonstrating a P value < 0.2 in the bivariate analysis were incorporated into a comprehensive multivariate logistic regression model. The final model was determined using a backward elimination strategy, with statistical significance established at a P value < 0.05. As the COVID-19 vaccination attitude scale appears to be a significant factor associated with vaccinations, we conducted additional analyses using chi-square tests to examine individual scale items as predictors of both COVID-19 vaccination and booster vaccination. All analyses were conducted using R [20].”

4. What ethical considerations were considered in the conduct of the study, particularly regarding the recruitment of participants from marginalized populations such as PWID? How were issues of informed consent, privacy, and confidentiality addressed throughout the research process?

Response: Thank you for your note. All participants involved in our study underwent a thorough informed consent process. Prior to their baseline interview appointment, participants were provided with detailed information about the study objectives, procedures, and the confidential nature of data collection. Our study was conducted in accordance with the guidelines and regulations of the New York University Institutional Review Board. Throughout the research process, study staff took extensive measures to safeguard the privacy of participants and the confidentiality of their data. We added a new subsection in our methods section to provide more details about ethical considerations in our study. The section reads:

“Ethical Considerations

Prior to the baseline interview appointment, all participants provided written informed consent after being fully informed about the study’s objectives, procedures, and the confidential nature of data collection. The informed consent process emphasized the voluntary nature of participation and the right to withdraw from the study at any time. Participation was confidential and all data collected were de-identified to prevent the identification of individuals. The study was approved by the New York University Institutional Review Board.”

5. The study identifies education level, vaccination attitudes, and drug-related characteristics as significant factors associated with booster vaccination uptake among PWID. Could you elaborate on the potential mechanisms underlying these associations and how they could inform targeted intervention strategies?

Response: Thank you for your comments. We revised the fourth paragraph of our discussion section to elaborate on the potential mechanisms underlying these associations and how they could inform targeted intervention strategies. It reads: 

“We found that individuals with lower education levels and anti-vaccine attitudes, particularly, towards the safety of the COVID-19 vaccine and its importance for health, were less likely to receive the COVID-19 booster vaccination. This could be explained by the fact that higher levels of education are often associated with a better understanding of the importance of vaccination, as consistent with findings in the general population in the US [22]. Attitudes towards vaccination can also be influenced by cultural beliefs, previous experiences with healthcare, trust in healthcare providers, and COVID-19 disinformation [2, 23]. These factors suggest that anti-vaccine attitudes play a role in influencing COVID-19 vaccine uptake. These findings are consistent with the existing literature on vaccine uptake among PWID and the US general population. For example, PWID in San Diego, holding the belief that COVID-19 vaccines contained a tracking device, reported lower COVID-19 vaccination rates, highlighting the role of disinformation spread through social media [2]. Another study conducted among PWID in San Diego and Tijuana, Mexico, also found a significant correlation between COVID-19-related disinformation and vaccine hesitancy in this population [23]. In a national study of US adults, conspiracy beliefs related to COVID-19 were associated with resistance to uptake of preventive behaviors and vaccination [24]. These findings underscore the necessity for targeted interventions to enhance vaccine trust and uptake in this marginalized population. Research indicates that attitudes can evolve, and interventions aimed at enhancing health literacy and debunking vaccine-related misconceptions could improve COVID-19 vaccine acceptance among PWID initially expressing hesitation [2, 3]. Additionally, community-based interventions involving peers and outreach efforts should prioritize debunking conspiracy theories and promoting health literacy among PWID [25].”

6. Based on the findings, what specific recommendations would the authors propose for public health interventions aimed at improving COVID-19 booster vaccination uptake among PWID in NYC? How might these recommendations be implemented in practice, considering the unique challenges faced by this population?

Response: Thank you for your comments. We revised the last paragraph of our discussion section to provide specific recommendations and their implementations in practice. It reads:

---

## [Editor Report · Decision Letter 1]

24 Apr 2024

COVID-19 vaccination uptake and determinants of booster vaccination among persons who inject drugs in New York City

PONE-D-24-02131R1

Dear Khezri Mehrdad,

We’re pleased to inform you that your manuscript has been judged scientifically suitable for publication and will be formally accepted for publication once it meets all outstanding technical requirements.

Kind regards,

Moses Katbi MD, MBA, MPH, FRSPH, DrPH

Academic Editor

PLOS ONE
---

## [Editor Report · Acceptance letter]

2 May 2024

PONE-D-24-02131R1 

PLOS ONE

Dear Dr. Khezri, 

I'm pleased to inform you that your manuscript has been deemed suitable for publication in PLOS ONE. Congratulations! Your manuscript is now being handed over to our production team.

Kind regards, 

on behalf of

Dr. Moses Katbi 

Academic Editor

PLOS ONE